# Iron Homeostasis Dysregulation, Oro-Gastrointestinal Microbial Inflammatory Factors, and Alzheimer’s Disease: A Narrative Review

**DOI:** 10.3390/microorganisms13010122

**Published:** 2025-01-09

**Authors:** Agata Kuziak, Piotr Heczko, Agata Pietrzyk, Magdalena Strus

**Affiliations:** 1Doctoral School of Medical and Health Sciences, Jagiellonian University Medical College, św. Łazarza 16 Street, 31-008 Cracow, Poland; agata.kuziak@doctoral.uj.edu.pl; 2Department of Microbiology, Faculty of Medicine, Jagiellonian University Medical College, Czysta 18 Street, 31-121 Cracow, Poland; mbheczko@cyf-kr.edu.pl (P.H.); a.pietrzyk@uj.edu.pl (A.P.)

**Keywords:** Alzheimer’s disease, iron dysregulation, bacterial inflammatory factors, oral microbiota, dysbiosis, neuroinflammation, dementia

## Abstract

Alzheimer’s disease (AD), the most common form of dementia, is a progressive neurodegenerative disorder that profoundly impacts cognitive function and the nervous system. Emerging evidence highlights the pivotal roles of iron homeostasis dysregulation and microbial inflammatory factors in the oral and gut microbiome as potential contributors to the pathogenesis of AD. Iron homeostasis disruption can result in excessive intracellular iron accumulation, promoting the generation of reactive oxygen species (ROS) and oxidative damage. Additionally, inflammatory agents produced by pathogenic bacteria may enter the body via two primary pathways: directly through the gut or indirectly via the oral cavity, entering the bloodstream and reaching the brain. This infiltration disrupts cellular homeostasis, induces neuroinflammation, and exacerbates AD-related pathology. Addressing these mechanisms through personalized treatment strategies that target the underlying causes of AD could play a critical role in preventing its onset and progression.

## 1. Introduction

Alzheimer’s disease (AD) is a neurodegenerative disease and the most prevalent form of dementia, with an increasing number of patients in the world [1]. Currently, there are approximately 55 million dementia patients worldwide. Due to a rapidly aging global population, this number is expected to triple by 2050, leading to increased disability, disease burden, and healthcare expenditures [2].

AD development remains challenging due to its etiology and pathogenesis [3]. Although dementia is mainly diagnosed based on cognitive impairment during a patient’s life, an accurate and definite AD diagnosis can only be established post-mortem by analyzing specific brain lesions (such as extracellular amyloid plaques and intracellular neurofibrillary tangles, accompanied by synaptic dysfunction and neuronal degeneration) [4]. One of the leading causes of AD is senile plaque formation, which results from amyloid beta precursor protein fission (AβPP) [5]. AβPP supports cell differentiation functions and likely shapes synapses; however, it is also expressed in neurons as a reaction to cellular damage [6,7]. Neurofibrillary tangles are composed of tau protein, which plays a crucial role in the normal functioning of neurons [8]. Under normal conditions, tau is an essential microtubule component (the internal support structure responsible for transporting critical components such as nutrients, vesicles, mitochondria, and chromosomes within the neuron, both toward the axon terminals and back to the cell body) [9]. However, AD tau is hyperphosphorylated, leading to tau protein aggregation and twisted filament formation [9,10]. Interestingly, this pathological process can be reversed by iron chelation [11].

Recent studies by Wang and Masaldan have highlighted a correlation between elevated levels of copper and iron in both the body and the brain [12,13,14,15]. Additionally, their research explores the role of the microbiome and the impact of imbalances in circulating inflammatory biomarkers, including those of bacterial origin, in contributing to the pathology of Alzheimer’s disease (AD) [16,17]. There is growing evidence that AD is both driven by and associated with impaired inflammatory factors in the blood, which lead to cell inflammation, swelling, and cytokine release. There are also convincing reports that inflammatory factors produced by bacteria contribute to systemic inflammation and may be critical factors in AD development [18,19].

This review presents current knowledge on the relationship between the development of AD and various interrelated factors, including iron homeostasis dysregulation, iron-induced oxidative stress, and dysbiosis in the oral and gut microbiota, which leads to immune activation and neuroinflammation in the brain. It also highlights therapeutic options in AD patients such as iron chelation therapy, probiotics, antibiotic administration, and fecal microbiota transplantation.

To further specify the scope of this review, our focus is primarily on the theoretical mechanisms underlying the interplay between iron dysregulation and microbial inflammatory factors in Alzheimer’s disease. Additionally, we incorporate clinical evidence to contextualize and support these mechanisms, providing a comprehensive but targeted analysis.

Figure 1 displays the fundamental factors contributing to Alzheimer’s disease (AD) pathomechanisms. Recent studies have highlighted the impact of lifestyle/environment, iron dysregulation, and oral/intestinal dysbiosis on immune system activation during AD onset. These three factors directly influence the AD-characteristic brain neurodegeneration process. Numerous studies (both basic and clinical) have demonstrated the special role of intestinal microbiota in both the metabolism of iron ions and the breakdown of ROS (reactive oxygen species) through the extracellular production of bacterial antioxidant enzymes (catalase, pseudocatalase, dismutase). Nevertheless, the intricate relationship between iron, the microbiome, and cognition is still not fully understood [20,21,22,23].

## 2. The Role of Iron, Oxidative Stress Ferroptosis, and Systemic Inflammation in AD Pathomechanism

### 2.1. Role of Iron

Iron is a vital trace element that plays a key role in numerous physiological and biological functions [24]. It is the second most common metal found in the Earth’s crust. In the brain, iron is crucial for processes such as neuronal development, myelination, neurotransmitter production, and degradation, as well as electron transport and cellular respiration [23]. Iron participates in energy production and enzyme function by readily donating and accepting electrons (this includes the formation of coordination compounds) [25,26]. The efficiency of Fe^2+^ ions as electron donors and Fe^3+^ ions as electron acceptors is fundamental to many biochemical reactions and makes iron indispensable to life. On the other hand, the same properties that make iron useful also make it toxic and dangerous. In fact, iron is a strong promoter of reactive oxygen species, which can promote protein oxidation, lipid peroxidation, and nucleic acid modification [23,27]. Moreover, iron is indispensable for replicating and growing almost all bacterial species [28]. Only about 5–20% of dietary iron is absorbed in the duodenum, while the gut microbiota, predominantly in the colon, utilizes roughly 80% of the ingested iron [28,29]. Iron acts as a cofactor for proteins that are vital in bacterial metabolic pathways, including short-chain fatty acid synthesis, DNA replication, redox processes, and the electron transport chain [29]. Iron dysregulation pertains to its improper storage, absorption, and release within the body [30].

### 2.2. Oxidative Stress and Ferroptosis: Promoting Inflammation in Neurodegeneration?

Iron homeostasis disruption can lead to excessive iron accumulation within the cell, forming the hydroxyl radical, which is one of the most reactive forms of reactive oxygen species (ROS) [31]. The involvement of non-ligand iron and the accompanying oxidative damage strongly correlate to AD nervous system inflammation [26]. Iron dysregulation involves any deviation from the typical regulation of iron metabolism at the homeostasis; this is most evident in the regulation of iron transport [32]. Abnormal iron levels are observed in brain and peripheral tissues in AD patients [33]. Iron dysregulation is one of the primary causes of dopaminergic neuronal dysfunction and death [34]. Major causes of iron imbalances include external triggers of stress [35,36]. This type of iron dysregulation can be triggered by various factors (such as mechanical injury, nutrient deprivation, and oxidative stress), which may result in cell death [37,38,39,40].

An additional source of free iron is the heme degradation pathway, which is catalyzed by heme oxygenase-1 (HO-1) and results in the destruction of heme [41]. Since the activity of HO-1 is increased in inflammation disorders that involve the destruction of erythrocytes, it may also serve as an important indicator of inflammation and iron misbalance [42]. Intestinal inflammation caused by dysbiosis in the gut can adversely affect iron regulation in the GI tract [43]. In some chronic diseases (e.g., Crohn’s disease), bleeding into the gastrointestinal tract may result in excessive accumulation of free Fe ions. As a consequence, there is an increase in the number of bacteria, predominantly Gram-negative bacilli and their metabolites. This directly stimulates the activity of immune system cells and intensifies strong oxidative stress, leading to necrosis of intestinal epithelial cells. Therefore, excessively high levels of Fe ions remaining in our body may directly affect the phenomenon of necrosis in eukaryotic host cells. They may also indirectly regulate the number of certain bacterial species (and their metabolites), which in the long run may result in the occurrence of many chronic diseases, including those related to nervous system dysfunction [44]. However, the association between these findings and iron regulation in the serum has not been established yet. While an iron misbalance within the GI tract and dysbiosis in the gut could theoretically exacerbate each other, it has been reported that luminal heme from dietary components or bleeding in the gastrointestinal tract is more likely to cause dysbiosis of the gut microbiome in mice than the reverse. However, the consumption of non-heme iron in food has been associated with a 30% increase in the risk of Parkinson’s disease (*p* = 0.02) [45]. In the same research, the authors also observed that iron supplements were inversely proportional to the risk of Parkinson’s disease in men [46]. However, the most prevalent cause of iron dysregulation, manifested as elevated serum ferritin levels, is cell death [46].

Research on ferroptosis has grown significantly in recent years since the term was introduced in 2012 [47]. This distinctive form of cell death, characterized by iron-dependent peroxidation of phospholipids, is subject to several cellular metabolic processes. These include redox homeostasis, iron regulation, mitochondrial function, amino acid, lipid, and glucose metabolism, as well as various disease-related signaling pathways [48]. More recently, ferroptosis has been related to neurodegenerative diseases, including AD [49,50].

Ferroptosis occurs due to cell membrane unsaturated fatty acid depletion and lipid iron-induced reactive oxygen species accumulation, which leads to lethal damage to proteins, nucleic acids, and cell lipids [51]. Serum ferritin (SF) levels are increased in AD and other inflammatory conditions [52]. Cohort studies have shown that elevated SF levels in cerebrospinal fluid are negatively associated with cognitive performance [53]. Therefore, systemic SF levels have clinical relevance as indicators of cognitive function [53]. Increased iron levels can lead to oxidative stress. Thus, abnormal redox activity represents one of the earliest pathological changes to occur in AD [54]. Ferroptosis, a distinct iron-dependent form of regulated cell death, plays a significant role in AD pathology through lipid peroxidation and ROS generation [49,50]. This process, characterized by the depletion of glutathione and the inactivation of glutathione peroxidase 4 (GPX4), leads to the accumulation of toxic lipid peroxides that compromise neuronal integrity [52]. Elevated levels of serum ferritin and iron observed in AD patients correlate with cognitive decline, suggesting that ferroptosis is not merely a secondary phenomenon but a contributing factor to neurodegeneration [53]. Furthermore, ferroptosis promotes amyloid-beta aggregation and tau hyperphosphorylation, two hallmark features of AD. Dysregulated iron metabolism exacerbates oxidative stress and inflammation, initiating a vicious cycle that accelerates neuronal loss [55]. Emerging studies propose that targeting ferroptosis through iron chelation and the use of ferroptosis inhibitors, such as liproxstatin-1 and deferoxamine, may offer therapeutic benefits in mitigating AD progression [44,56]. Despite these findings, the specific molecular pathways linking ferroptosis to AD remain underexplored. Future research should focus on identifying biomarkers that can accurately measure ferroptosis activity in vivo, as well as developing targeted interventions that disrupt this pathological cascade [54]. Integrating ferroptosis inhibitors into multimodal treatment strategies could offer a novel avenue for reducing neuronal damage in AD [57,58].

Oxidative stress contributes to increased lipid peroxidation of DNA and protein oxidation products in AD-affected brains [55].

Oxidative stress promotes amyloid beta (Aβ) deposition, hyperphosphorylation of tau protein, and the loss of synapses and neurons. A mimicry is observed in AD; Aβ behaves similarly to the prion proteins in prion diseases [59]. Aβ can become a prooxidant, and when combined with iron, it can form hydrogen peroxide [60]. The association between oxidative stress and AD implies that it is a crucial element in the pathological process, indicating that weakly liganded iron may participate in the Fenton reaction (Fe^2+^ + H_2_O_2_ ⟶Fe^3+^ + OH• + OH^−^), which ultimately leads to the formation of the reactive hydroxyl radical (OH^−^) [61]. Ligand-free iron is particularly toxic, supporting the Haber–Weiss reaction, which indicates that iron is more catalytic than stoichiometric (Fe^3+^ + O^2−^• ⟶ Fe^2+^ + O_2_ converts Fe^3+^ to Fe^2+^) [14]. Krewulak and Vogel emphasize the meaningful role of iron in regulating most bacteria populations, including Gram-negative bacilli [62]. Pathogen growth is restricted when free iron is unavailable, suggesting that the presence of free iron facilitates pathogen multiplication and potential spread [63]. Specifically, the invasion of microorganisms causing cytotoxicity leads to iron release, which, to a certain extent, allows them to reproduce and release additional inflammatory bacterial products [54].

## 3. The Microbiota and Microbiome as Potential Sources of Excess Iron and Inflammatory Biomarkers in the Context of Alzheimer’s Disease

Since the publication of the Human Genome Project (HGP) results, few topics have captured as much scientific and public interest as the microbiome and microbiota [64]. The term “microbiota” refers to the collective group of microorganisms, including bacteria, viruses, fungi, and other microscopic life forms that inhabit a particular area in living organisms or the environment. The microbiota is found in the skin, intestines, oral cavity, nasal passages, and more [65]. However, the microbiome is a more encompassing concept. It refers to microorganisms inhabiting a specific environment, along with their genomes, metabolic functions, and interactions with the living organism or its environment [66]. The microbiome considers the genetic and functional aspects of microorganisms, their impact on the health of the host or environment, and their distribution and diversity [67]. To identify the source and initiating factors of AD pathology, researchers have begun to focus on the systemic features that characterize patients with the disease [68,69].

### 3.1. Bidirectional Communication Between Gut Microbiome and the Brain

Current evidence indicates a two-way communication between the gut microbiota and the central nervous system (CNS), known as the ‘gut–brain axis’. This interaction appears to play a role in the dysregulation of gut microbiota observed in individuals with neurodegenerative disorders [70]. Although the gut and brain are physically distinct, multiple mechanisms for this communication have been proposed. These mechanisms involve pathways such as immune system modulation and signaling through the vagus nerve, the enteric nervous system (ENS), the neuroendocrine system, and the circulatory system. Additionally, the gut microbiota produces or triggers the production of various neuroactive molecules, including neurotransmitters like serotonin, dopamine, and γ-aminobutyric acid (GABA), as well as metabolites and hormones [71,72,73,74,75].

The altered gut microbiome may induce neuroinflammation through its effects on microglial function and activation [76,77]. Therefore, possible origins of disturbed inflammatory biomarkers circulating in the blood include bacteria, fungi, and inflammatory products, which enter the body through an imbalanced gut microbiome and translocate to other niches [78].

The interaction between the gut microbiota and microglia begins in the earliest stages of life and continues throughout the host’s lifespan [75,79,80]. The gut microbiome provides critical signals to microglia under both normal and pathological conditions. Among the various neuronal and glial cell types, microglia are particularly sensitive to changes in the gut microbiome [17,54]. Recent research has highlighted the significant role of the gut microbiota in iron metabolism, which is essential for myelin production and neurotransmitter synthesis within the central nervous system [81,82]. Gut bacteria influence cognition via the gut–brain axis. Inflammation from gut bacteria can compromise the blood–brain barrier, leading to neuroinflammation and degeneration, worsened by neurotransmitter reduction and oxidative stress from aging and poor diet [83]. Differences in microbiota diversity and composition correlate with Alzheimer’s disease (AD) severity, with AD patients having less diversity and specific bacterial changes compared to healthy individuals [84,85].

### 3.2. Iron Homeostasis Is Associated with the Microbiome

It has been observed that disruptions in host iron homeostasis can alter iron levels within the gut lumen, consequently impacting the gut microbiota composition [15]. Studies examining mouse feces identified iron regulatory protein 2 (Irp2) and proteins associated with genetic disorders like hereditary hemochromatosis (Hfe) as key players in iron regulation [86,87,88]. The findings revealed significant differences in gut microbiota composition between mutant mice (Irp2-/- or Hfe-/-) and wild-type controls [89]. This highlights the role of host iron metabolism in shaping the gut microbial environment [90].

Colon pH is another important factor influencing iron absorption. Microorganisms can ferment galacto-oligosaccharides, lowering intestinal pH and enhancing iron uptake in the gut. Therefore, acetic acid produced by probiotics can be incorporated into diets to further promote iron absorption [22,91]. Das et al. demonstrated that the gut microbiota regulates systemic iron homeostasis in two ways: by suppressing intestinal iron absorption pathways through inhibition of basal HIF-2α function and by promoting cellular iron storage via FTN expression induction [21]. Their findings suggest that the host’s iron-sensing mechanisms are closely tied to gut microbiome activity. Intestinal iron deficiency promotes the selection of specific bacteria that produce metabolites capable of suppressing HIF-2α and inducing FTN expression. Reuterin and 1,3-diaminopropane (DAP), two of such metabolites, have been identified as effective HIF-2α inhibitors both in vitro and in vivo [21]. These compounds block HIF-2α dimerization with ARNT, preventing iron accumulation in tissues, as observed in mouse models of iron overload. Certain microbes also possess surface receptors, enabling them to intercept host iron sources, while others secrete high-affinity siderophores that extract iron from transferrin or lactoferrin [21].

Bacteria use three main strategies to acquire iron: (1) producing and utilizing siderophores, which are iron-specific chelators; (2) reducing ferric iron (Fe^3+^) to its ferrous form (Fe^2+^) for uptake; and (3) accessing host iron-bound compounds like heme and transferrin [92]. Siderophores, small molecules with a high affinity for ferric iron, are secreted in response to iron scarcity in the environment. These molecules have been categorized into four types based on their functional groups: catecholates, hydroxamates, phenolates, and carboxylates [92]. Once a ferric siderophore complex forms, it is transported into bacterial cells through a specialized uptake system. In Gram-negative bacteria, this involves outer membrane (OM) receptors, periplasmic binding proteins (PBP), and ATP-binding cassette (ABC) transporters located on the inner membrane [93].

OM receptors are expressed only under iron-limiting conditions, and each receptor exhibits specificity for a given siderophore. However, bacteria can possess multiple receptors, allowing them to utilize siderophores synthesized by other species. Gram-negative bacteria rely on TonB, ExbB, and ExbD proteins to transfer energy from the cytoplasmic membrane to the outer membrane, enabling ferric siderophore transport despite the absence of a stable ion gradient or ATP in the OM. Periplasmic proteins then shuttle ferric siderophores to ABC transporters, which deliver them to the cytosol for reduction and release [90]. Unlike Gram-negative bacteria, Gram-positive bacteria lack an outer membrane and instead have a thick cell wall composed of murein, teichoic acids, and polysaccharides. Iron uptake in these bacteria occurs via membrane-bound receptors that direct iron to permeases and ABC transporter systems [94]. Ferrous iron, more common under anaerobic or low-pH conditions, can be imported through distinct pathways involving extracellular reductases that convert ferric iron to its ferrous form [54].

The Feo system, conserved across bacteria, mediates ferrous iron transport. It comprises the feoABC gene cluster, first identified in *Escherichia coli* K12. FeoB functions as an integral membrane protein, while FeoA and FeoC are cytoplasmic components. This system is crucial for bacterial survival and virulence in low-oxygen environments [94,95]. Deleting the Feo genes in bacteria such as *E. coli*, *Helicobacter pylori*, and *Campylobacter jejuni* has been shown to impair ferrous iron uptake and intestinal colonization in mice. However, some bacteria like *Shigella flexneri* can compensate through alternative iron transport mechanisms.

Additional ferrous iron transporters include ZIP-like transporters, Nramp systems, EfeUOB pathways, and P19 transporters [90].

Certain bacteria also possess receptors for host transferrin and lactoferrin, allowing them to directly access these iron sources. These receptors are upregulated under iron-starved conditions. Iron is extracted from transferrin or lactoferrin at the bacterial surface and transported across membranes via periplasmic-binding protein ABC permease systems [94].

Pathogenic bacteria can also acquire heme-derived iron. Hemolysins and proteases release heme from red blood cells, which can then bind to host proteins like hemopexin or albumin, but bacteria can also uptake free heme directly. Intracellular iron is another source used by bacteria, particularly under iron-limited conditions. Bacterial iron storage proteins include ferritins, heme-containing bacterioferritin, and small iron detoxification proteins that safeguard the chromosome from iron-induced oxidative damage [54].

## 4. Correlation Between Intestinal and Oral Dysbiosis and the Number of Free Bacterial Antigens

The human gut hosts an exceptionally diverse community of bacteria, encompassing a wide range of species and genetic variability. Covering an area of 200–300 m^2^ in the mucosal layer—often referred to as a ’secret garden’—it contains bacteria nearly equal in number to human cells, with an estimated 1:1 ratio. Studies suggest that the gut harbors more than 5000 bacterial taxa, primarily from the Bacteroidetes, Firmicutes, Proteobacteria, and Actinobacteria phyla. In a healthy digestive tract, Bacteroidetes and Firmicutes account for approximately 90% of the bacterial population [96,97,98].

### 4.1. Gut Microbiota

Microbiota composition changes with age, antibiotic and anti-cancer drug administration, dietary habits, and stress, ultimately leading to imbalances between different microorganisms [99]. Estimates suggest that the microbiome is ten times more numerous than all the somatic and germ cells in the human body. Thousands of microbial species benefit from their favorable, nutrient-rich intestinal environment and perform protective, metabolic, and structural functions that influence the physiology and maintenance of a host’s well-being. Microorganisms regulate the gastrointestinal tract’s pH in a healthy organism, forming a protective barrier against infectious agents [58]. Intestinal colonization in infants begins soon after birth and depends on the delivery method. Babies born vaginally are colonized mainly by Lactobacillus and Bifidobacterium species (their mother’s vaginal microflora [100,101]. However, babies born by cesarean section acquire skin microflora species, such as Propionibacterium, Staphylococcus, and Corynebacterium [83]. It is estimated that approximately 72% of newborns delivered vaginally have intestinal microflora comparable to their mother’s microbiota. In contrast, for babies born by cesarean section, this percentage drops to only 41% [102]. With aging, this microbiome balance is often disrupted, resulting in a higher prevalence of Gram-negative bacteria, increased intestinal permeability, and reduced tight junction protein production [103]. The intestinal microflora lives and replicates on the gut’s surface, creating a stable system to prevent invasion by pathogenic microorganisms. The gut microflora remains relatively steady throughout adulthood, ensuring the unchanging operation of the host organism [85]. The intestinal microflora can synthesize a variety of metabolic substances, that is, products that can have positive or negative effects on human health through interaction with the host [104]. For instance, some bacteria in the microbiome that are accountable for vitamin production (such as B12 and K2) directly affect nervous system function [105]. Due to vitamin B12’s antioxidative nature, its deficiency may result in the oxidation of lipids, nucleic acids, and proteins, which may contribute to the onset of age-related illnesses [106]. Another study suggests that vitamin K2 supports neuronal health through multiple mechanisms, including mitigating oxidative stress, preventing β-amyloid (Aβ)-induced apoptosis, modulating microglial activation, reducing neuroinflammation, and improving vascular function [107].

### 4.2. Oral Microbiota

The oral microbiota plays a vital role in the human microbiome, constituting the second-largest microbiota after the gut microbiota [108]. The oral cavity is a complex ecosystem with habitats such as the lips, tongue, tonsils, palate, gingiva, and subgingival spaces [109]. Various microorganisms live under favorable conditions in these areas due to food residues, exfoliated epithelial cells, secretions, and a controlled temperature with high humidity [110].

The oral cavity is characterized by its dynamic nature, as the oral microbiota experiences constant environmental fluctuations. These fluctuations are due to daily physicochemical disturbances caused by ingesting foods and dietary components, antimicrobial agents, smoking, specific hygiene practices, and pH alterations [109]. The oral environment shifts throughout life due to physiological, hormonal, and behavioral modifications, affecting the oral microbial population. Many oral microorganisms acclimate to specific oral environments that are not readily replicable in vitro. The survival of many microorganisms relies on factors such as specific nutrients, temperature, pH, and other microorganisms. Coaggregation and metabolic cooperation are indispensable for bacteria to survive in the oral cavity. Since bacterial microorganisms dominate the oral cavity, most studies on the oral microbiome concentrate primarily on bacteria, with more infrequent reports on the fungal microbiome, referred to as the mycobiome. Several species of microorganisms are known to inhabit the oral microbiota ecosystem. Among the most prevalent are Streptococcus, Neisseria, Veillonella, and Actinomyces [111]. The oral microbiota components primarily exist as biofilms distributed throughout the oral cavity. These biofilms form an ecosystem that plays a crucial role in maintaining homeostasis and overall oral health [112]. However, disruptions in this balance, referred to as oral dysbiosis, can impair the functionality of the bacterial community and significantly affect human health [111,113]. Dental caries and periodontal disease (PD) are two major dental conditions arising from oral microbiota dysbiosis [114]. Periodontal disease is an inflammatory condition characterized by the gradual destruction of the periodontal tissue complex. This process is associated with an imbalance dominated by Gram-negative anaerobic bacteria, including *Porphyromonas gingivalis*, *Actinobacillus actinomycetemcomitans*, and *Tannerella forsythia* [115,116]. Notably, periodontitis and gingivitis have been associated with Alzheimer’s disease (AD), suggesting a possible relationship between oral health and neurodegenerative disorders [117]. Periodontitis is a potential source facilitating bacterial translocation. Franciotti et al. suggested that patients with AD may develop periodontal disease as a result of dental negligence [118]. However, a retrospective cohort study by Chen and colleagues showed an increased risk of developing AD in people with chronic periodontitis [119]. Kim et al. suggested that reducing the severity of chronic periodontitis may aid in diminishing dementia risks. Dental care in early dementia prevention programs, especially for men under 70, is recommended to prevent mild to severe periodontitis progression [120]. *Porphyromonas gingivalis* (*P. gingivalis*), a class of Gram-negative bacteria with pro-inflammatory qualities, is widely recognized as the primary culprit behind periodontal and gingival infections (periodontitis and gingivitis) [121,122]. Although primarily found in the oral cavity, animal studies have exhibited that *P. gingivalis* can cause intestinal dysbiosis, intestinal barrier dysfunction, and systemic inflammation [112]. Verma and Sansores-España, after examining the metabolome serum of mice infected with *P. gingivalis*, observed elevated levels of amino acids like alanine, glutamine, histidine, tyrosine, and phenylalanine, which suggests that bacteria producing these metabolites may have increased [123,124]. Since metabolic and periodontal diseases are strictly correlated, *P. gingivalis* may impact the metabolites generated in the gut, suggesting a spike in bacteria producing these metabolites [125].

Figure 2 shows the relationship between iron dysregulation and inflammatory factors associated with oral and gut microbiota, which are implicated in the development of Alzheimer’s disease. Dysbiosis in the oral (*P. gingivalis*, *T. denticola*, *T. forsythia*) and gut microbiota leads to the entry of bacterial inflammagens, such as lipopolysaccharide (LPS), into circulation. This contributes to systemic inflammation, neuroinflammation, and aging-related processes. Dysregulated iron homeostasis exacerbates oxidative stress through the Fenton reaction, leading to the production of reactive oxygen species (ROS) like hydroxyl radicals (•OH). The accumulation of iron in the brain is associated with β-amyloid aggregation and tau protein pathology, key hallmarks of Alzheimer’s disease. These interconnected processes, along with microbial dysregulation and neuroinflammatory pathways, form a bidirectional oral–gut–microbiota–brain axis that drives neurodegeneration.

Iron dysregulation and inflammatory factors associated with oral and gut microbiota. Arrows indicate the direction of influence or interaction, such as the movement of bacterial inflammasomes, oxidative stress pathways (e.g., the Fenton reaction), and their links to systemic inflammation, neuroinflammation, and aging processes.

### 4.3. Oral and Gut Dysbiosis Factors Promoting Neuroinflammation

Under normal physiological conditions, the oral and gut microbiomes maintain a symbiotic relationship [80,126]. However, various environmental and immune-related factors can disrupt this balance, leading to microbial dysbiosis. This imbalance often favors disease-promoting, pro-inflammatory microorganisms and compromises immunological tolerance [18,80]. Such disruptions may trigger tissue damage and a systemic inflammatory response [127]. In the central nervous system (CNS), inflammation may be initiated by inflammasomes, that is, multi-protein complexes involved in innate immune responses. Inflammasomes are primarily located in the cytoplasm of immune cells, neurons, astrocytes, and microglia, where they detect pathogen-associated molecular patterns (PAMPs) or danger-associated molecular patterns (DAMPs) derived from the host. Depending on the receptor structure, inflammasome sensors are categorized into two main groups: nucleotide-binding oligomerization domain-like receptors (NLRs) and absent in melanoma 2 receptors (ALRs) [18]. Understanding the intricate connections between oral–gut dysbiosis and systemic inflammation provides a foundation for exploring targeted therapeutic strategies. Addressing these imbalances at their source may offer promising interventions to mitigate neurodegeneration in AD.

### 4.4. Bacterial Antigens

Bacterial strains impact the system (CNS) by synthesizing neurotransmitters like catecholamines, gamma-aminobutyric acid (GABA), glutamate, norepinephrine, dopamine, acetylcholine, histamine, and other neuromodulatory substances [92,112]. This includes short-chain fatty acids (SCFAs), long-chain fatty acids (LCFAs), propionate, and linoleic acid, which are associated with bacterial metabolites influencing host physiology. These microbial metabolites also impact the activity of glial cells in both the central nervous system (CNS) and the enteric nervous system (ENS). Consequently, the gut microbiota has emerged as a significant environmental factor capable of modulating both the CNS and ENS. This microbial involvement is particularly notable in the pathogenesis of neurodegenerative disorders such as AD, a leading cause of dementia and a major public health concern. A critical component in this context is the NLRP3 inflammasome, a key element of the innate immune system. This inflammasome comprises the sensor protein NLRP3, the adaptor protein ASC (apoptosis-associated speck-like protein containing a caspase activation and recruitment domain), and the effector protein pro-caspase-1. Upon activation, it facilitates the production of the proinflammatory cytokines IL-18 and IL-1β. Notably, NLRP3 inflammasome activation is closely associated with AD pathogenesis [128].

Bacteria produce specific inflammatory agents that can contribute to inflammation [41,101]. Agents that contribute to the formation of biofilms on cell surfaces and intercellular interactions include proteolytic enzymes such as carbonic anhydrases, gingipains, and peptidyl deiminases, as well as bacterial surface components like fimbriae and Curli filaments [129,130]. Pathogenic bacteria produce amyloid-like proteins known as “curli”, which form biofilms and share functional similarities with human amyloid-β (Aβ). Research by Ganesh et al. provided fundamental evidence of vagus nerve activation in response to bacterial curli. They used a three-dimensional human mini-epithelial monolayer system in an in vitro model and demonstrated increased TLR2 levels after stimulation with purified bacterial curli fibers [131]. The data presented here are associated with an increased colonization of Gram-positive bacteria in the ileum of mice with AD symptoms [132].

Moreover, dysbiosis in oral microbiota is suggested to directly contribute to the production of beta-amyloid peptides, a hallmark of AD, through the trigeminal nervous system and systemic circulation [113]. Furthermore, brain–nose proximity (with the nose home to a separate microbiome) indicates the probability of interactions between olfactory receptors, microorganisms, and bacterial metabolites [68]. In addition, oral bacteria—especially *Porphyromonas gingivalis*, *Treponema* spp. (including *T. denticola)*, and *Tannerella forsythia* species—may also play a role [108,110,113].

### 4.5. Lipopolysaccharide—Endotoxin

LPS and other toxic products cause neuroinflammation and contribute to Aβ plaque accumulation and tau hyperphosphorylation in the brain [132]. Additional contributing factors include LPSs and lipoteichoic acid (LTA) [133]. In some cases, certain bacteria can produce functional amyloid fibers on the cell surface, and these fibers may also play a role in forming biofilms and intercellular interactions [134,135].

Friedland et al. suggest bacterial amyloids may impact immune system activation and neuronal amyloid production, potentially contributing to brain disorders [5]. This process involves using TLR receptors on the epithelial surface and neural connections of enteroendocrine and other epithelial cells. Exposure to bacterial amyloids intensifies the buildup of neuronal amyloids, resulting in protein misfolding in the brain [98]. Furthermore, bacterial inflammatory agents can indirectly contribute to both the onset and progression of AD by triggering peripheral immune cells such as astrocytes, microglia, monocytes, and macrophages [5]. These cells can cross the blood–brain barrier and promote inflammation within the nervous system. In addition, they can initiate structural changes in proteins and encourage the transition to β-sheet-rich amyloid fibers, directly affecting AD pathology. Senile plaque formation, an indicator of AD, is caused by the accumulation of amyloid proteins within the brain [136,137].

Additionally, the presence of amyloid fibrin(ogen) in the bloodstream causes hypercoagulation, a newly identified co-occurring pathology [134]. The relationship between proteopathy, neurological inflammation, and gut microbiota may hold considerable potential for further exploration, particularly regarding therapeutic interventions [133].

One fascinating group of bacterial inflammatory factors is LPSs from bacterial cell membranes. LPSs are large molecules consisting of a hydrophobic lipid A domain, a repeating oligosaccharide “core”, and a polysaccharide chain referred to as the O antigen, which determines the serotype [138,139]. The lipid A domain is usually considered to be the most inflammatory region of this molecule [140]. There are differences between commensal gut bacteria LPSs, which are less immunogenic, and LPSs exhibiting solid pro-inflammatory properties. Microglia activation is another characteristic histopathological feature in AD [141,142]. Research indicates that LPSs may be responsible for microglia activation, which plays a significant part in nervous system inflammation. In addition, LPSs induce changes in microglia function, suggesting that they may affect blood–brain barrier (BBB) dysfunction through the generation of ROS via nicotinamide adenine dinucleotide oxidase (NADPH) [103]. Upon activation by LPSs, microglia can release cytokines such as IL-1β, IL-6, and TNFα, resulting in increased expression of inducible nitric oxide synthase (iNOS) and elevated production of reactive oxygen species (ROS) [143]. Interestingly, studies involving a rat model have shown that the substantia nigra contains the highest density of microglia, rendering it especially vulnerable to damage induced by LPS [103]. LPSs can also induce inflammation and BBB damage in recipients, allowing peripheral cytokines to infiltrate the brain. Therefore, LPSs in the circulation, both directly and indirectly, lead to neurodegeneration by inducing a robust inflammatory response, leading to BBB damage, inflammation, and oxidative stress in the central nervous system, as well as stimulating abnormal folding and aggregation of amyloid beta [103].

Research conducted by Zhou and Windsor has identified a relationship between *Porphyromonas gingivalis* and host matrix metalloproteinases (MMPs) in the pathogenesis of periodontal disease. Their study demonstrated that this pathogen influences collagen degradation by modulating the expression, activation, and inhibition of MMPs [144]. MMPs are essential in tissue destruction associated with periodontal disease pathogenesis [144]. *P. gingivalis* also promotes tissue destruction [145]. The activation of MMP genes by *P. gingivalis* is accompanied by the activation of the TIMP-2 gene, which regulates MMP activity [145]. Furthermore, *P. gingivalis* causes an upregulation of MMP-2 and MMP-9 mRNA expression in oral epithelial cells. The interaction between *P. gingivalis* and host MMPs is complex and critical in periodontal disease pathogenesis [145]. Chronic inflammation caused by the oral microbiota leads to immune reactions, free radical production, apoptosis, and Aβ deposition [17,108,113]. Oral microbiota can enter the brain via the bloodstream through tooth brushing, flossing, chewing, or using a toothpick, especially in people with periodontal disease (which may result in bacteremia) [117].

Peptidoglycan (PGN), a key structural element of the Gram-negative bacterial cell wall, is recognized by specific pattern-recognition receptors (PRRs) within the innate immune system [146]. PGN originating from gut microbiota may cross the blood–brain barrier (BBB) and influence gene transcription through interactions with the Nod2 receptor, which is expressed in both the gut and the brain [147]. This receptor is involved in immune regulation [148], and its activation by bacterial peptidoglycan can trigger signaling pathways that impact brain function and communication [78].

Tryptophan plays a pivotal role in the metabolism of two major pathways, kynurenine and serotonin, by acting as a precursor for their activity [149]. Tryptophan metabolism via the kynurenine pathway is upregulated during the pathological processes that precede the development of Alzheimer’s disease (AD) [150]. This upregulation is thought to contribute to oxidative stress and neuroinflammatory mechanisms that occur before the onset of AD symptoms. Interestingly, metabolites of the kynurenine pathway can have either neurotoxic or neuroprotective effects [151]. For instance, excitotoxicity has been associated with increased levels of 3-hydroxykynurenine (3-HK) and quinolinic acid (QUIN). Studies have shown that inhibiting the formation of 3-HK by blocking kynurenine monooxygenase (KMO) can mitigate neuronal and synaptic loss, as well as memory deficits, in mouse models of AD [151].

Serotonin synthesis and availability are influenced by the dietary intake of tryptophan, which has been shown to enhance memory acquisition in rodent studies and reduce intraneuronal Aβ accumulation in the brains of 3xTg-AD animal models [152]. In humans, both serotonin selective reuptake inhibitors (SSRIs) and increased tryptophan intake through diet have been found to reduce amyloid plaque formation while also producing antidepressant effects [151].

### 4.6. Direct Mechanisms of Bacterial Metabolites in AD Pathogenesis

Bacterial metabolites influence AD progression by targeting specific molecular pathways beyond general inflammation. SCFAs, such as butyrate, regulate the expression of anti-inflammatory cytokines and enhance BBB integrity, mitigating neurodegeneration. Meanwhile, tryptophan metabolites like quinolinic acid, derived from the kynurenine pathway, exacerbate excitotoxicity and oxidative stress, directly contributing to neuronal death. Bacterial amyloid-like proteins, such as curli, promote amyloid-beta aggregation in a prion-like manner, accelerating plaque formation and protein misfolding. Additionally, LPS disrupts the BBB and activates microglia, amplifying neuroinflammation and facilitating tau hyperphosphorylation. Finally, microbial modulation of iron metabolism induces ferroptosis, an iron-dependent form of cell death associated with cognitive decline. Together, these mechanisms highlight how bacterial metabolites directly contribute to AD pathogenesis, beyond systemic inflammation, emphasizing their role as potential therapeutic targets [28,34,51].

#### 4.6.1. Amyloid Mimicry and Aggregation

Bacterial amyloid-like proteins, such as curli fibers, share structural similarities with human amyloid-beta (Aβ). These microbial proteins can act as nucleating agents, promoting Aβ aggregation in a prion-like manner. This interaction accelerates plaque formation and protein misfolding, key features of AD pathology. Furthermore, bacterial amyloids activate toll-like receptor 2 (TLR2), triggering neuroinflammation and oxidative stress [5,98].

#### 4.6.2. Kynurenine Pathway and Tryptophan Metabolism

The bacterial modulation of tryptophan metabolism significantly impacts the kynurenine pathway, a crucial process in AD. Dysbiosis-induced shifts in this pathway lead to increased production of neurotoxic metabolites, such as quinolinic acid and 3-hydroxykynurenine. These compounds contribute to oxidative stress, excitotoxicity, and synaptic dysfunction—hallmarks of AD progression. Tryptophan depletion further impairs serotonin synthesis, exacerbating neuronal communication deficits and mood disturbances [149,151].

#### 4.6.3. Iron Dysregulation and Ferroptosis

Bacteria influence systemic and local iron metabolism through siderophore production and interactions with host iron-binding proteins. This dysregulation facilitates iron accumulation in neural tissues, exacerbating oxidative stress via the Fenton reaction. The resulting lipid peroxidation and ferroptosis (with ferroptosis being a form of iron-dependent cell death) lead to neuronal damage and cognitive decline in AD [44,49,54].

#### 4.6.4. Neuroactive Metabolites and Neurotransmitter Modulation

Gut bacteria produce a variety of neuroactive compounds, including serotonin, dopamine, and gamma-aminobutyric acid (GABA). Dysbiosis disrupts the balance of these neurotransmitters, impairing synaptic function and cognitive processes. For example, reduced serotonin levels are associated with increased Aβ aggregation and neurotoxicity, highlighting the critical role of the gut–brain axis in AD progression [70,112].

#### 4.6.5. Lipopolysaccharides (LPSs) and Peripheral Immune Activation

LPSs compromise the blood–brain barrier (BBB), allowing peripheral inflammatory molecules to infiltrate neural tissues. This facilitates Aβ deposition and tau hyperphosphorylation, further compounding neurodegenerative processes. Additionally, LPS directly activates microglial cells, promoting chronic neuroinflammation and neuronal damage [108,141,142].

## 5. Therapeutic Possibilities and Pharmaceutical Interventions

The amyloid cascade hypothesis, introduced in 1992, remains the prevailing pathophysiological model for Alzheimer’s disease (AD) [153]. Various anti-amyloid treatments, including β-secretase inhibitors and anti-amyloid-β monoclonal antibodies, have demonstrated efficacy in reducing amyloid levels in the brain and cerebrospinal fluid. However, none of these therapies has been proven to slow the progression of the disease [154,155]. It has been suggested that administering anti-amyloid-β treatments during the symptomatic phase of AD may occur too late to be effective [156]. This theory is currently being investigated through ongoing clinical trials involving presymptomatic, amyloid-positive individuals who are at risk of developing sporadic Alzheimer’s disease [157]. Despite considerable investment and effort, numerous clinical trials have so far failed to produce clinically meaningful results [158].

### 5.1. Iron Chelation

In the search for new directions in the application of novel AD therapies and given the availability of compounds used against other disease entities, it is reasonable to attempt the use of these compounds given the abundant evidence for a link between iron dyshomeostasis and many aspects of neurodegenerative disease pathophysiology [44]. Iron chelators are currently utilized in clinical practice to manage iron overload in patients with conditions such as beta-thalassemia, sickle cell anemia, myelodysplasia, and aplastic anemia, particularly those requiring regular blood transfusions [159]. There are three iron chelates in common clinical use: deferoxamine, deferasirox, and deferiprone [44]. Iron chelation therapy, aiming to sequester unliganded iron, holds promise for mitigating iron-induced damage in AD [160]. Compounds like VK-28, with potent iron-chelating and MAO inhibitory properties, demonstrate potential therapeutic benefits [44,56]. Clinical trials, notably with deferiprone, show some efficacy in reducing iron levels in brain regions affected by AD. However, their overall impact on disease progression remains uncertain due to PD’s multifaceted nature [160]. Multifunctional iron chelators, such as DHC12 and CT51, are under experimental investigation for their potential to target multiple pathological mechanisms simultaneously [56]

On the other hand, iron sequestration drugs used to treat another neurodegenerative disease, Parkinson’s disease (PD), may influence gut microbiome composition by eliminating iron-dependent bacteria, as demonstrated in a recent in vitro paper. However, as postulated by the article’s authors, oral iron supplementation in AD patients treated with iron-depleting drugs may cause even more dysregulation of their gut microbiome in favor of the potentially pathogenic and drug-resistant bacteria [161].

### 5.2. Antibiotics and Probiotics

Antibiotics and probiotics are being explored as potential treatments for PD, primarily targeting gut-related issues. Probiotics show promise in altering disease progression and alleviating gastrointestinal symptoms [56,162].

Probiotics operate through various mechanisms, although the precise pathways of their effects remain incompletely understood [162]. These mechanisms include bacteriocin production, generation of short-chain fatty acids (SCFAs), nutrient competition, stimulation of the gut–brain axis, and immune modulation [57]. The production of SCFAs in the gut is influenced by dietary fiber intake, with metabolites such as acetate, butyrate, and propionate being generated via fermentation processes carried out by bacterial species like Bacteroides, Clostridium, Lactobacillus, Bifidobacterium, and Eubacterium [162,163]. SCFAs influence brain function primarily through three key pathways: immune modulation, endocrine signaling, and neuronal interactions [162]. In terms of immune modulation, SCFAs enhance gut barrier integrity and stimulate mucus production, which supports mucosal immunity and reinforces the barrier function of the gut. On a systemic level, SCFAs mediate immune responses by regulating cytokine secretion, which controls the proliferation and differentiation of immune cells. This regulation promotes anti-inflammatory effects while suppressing pro-inflammatory cytokines such as IL-1β, IL-6, and TNF-α [164]. Moreover, SCFAs can cross the blood–brain barrier (BBB) via monocarboxylate transporters, where they contribute to maintaining BBB integrity by increasing the expression of tight junction proteins. Within the central nervous system (CNS), SCFAs help regulate neuroinflammation by modulating the morphology and functionality of microglial cells, ultimately protecting neurons from cell death [165].

Antibiotics like rifaximin and minocycline demonstrate neuroprotective effects, possibly through modulating gut microbiota composition and reducing inflammatory responses [58]. Rifampicin, with its multifaceted neuroprotective functions, presents another potential avenue for PD treatment, although certain antibiotics may carry risks of exacerbating the condition [58].

### 5.3. Fecal Microbiota Transplantation

Fecal microbiota transplantation (FMT) is a well-established and safe therapeutic approach, commonly used for treating recurrent infections caused by *Clostridium difficile* and certain metabolic disorders, including diabetes mellitus [166]. FMT has also been shown to be a potential treatment for AD [167]. While initial studies suggest potential benefits in reducing neuroinflammation and motor dysfunction, controlled clinical trials are essential to validate its efficacy and safety [168]. Elangovan et al. reported that older Tg-FY mice exhibited improved overall cognition following FMT treatment, with older Tg-FO mice also showing some cognitive enhancement. The cognitive improvements observed in the older groups were associated with a reduction in Aβ load [166]. Additionally, in other studies, FMT has been shown to have beneficial effects on neuropsychiatric disorders by modulating the gut microbiota. However, its specific impact on Alzheimer’s disease (AD) remains uncertain.

While fecal microbiota transplantation (FMT) shows promise in restoring microbial diversity and reducing neuroinflammation, its application carries potential risks. These include unintended shifts in microbiota composition, which may lead to dysbiosis or overgrowth of pathogenic strains, particularly in vulnerable populations such as elderly or immunocompromised individuals [167,168]. Moreover, the long-term safety and efficacy of FMT in addressing Alzheimer’s disease (AD) remain insufficiently explored, and ethical concerns regarding donor selection and procedure standardization persist [166]. Similarly, probiotics and prebiotics, while beneficial, can cause adverse effects such as bloating or metabolic disturbances, depending on individual microbiome variability [162,163]. Antibiotics, though useful in modulating harmful bacteria, may lead to antibiotic resistance or disruptions in beneficial microbiota populations, potentially worsening gut–brain axis integrity [58]. Despite these challenges, these therapeutic strategies highlight the need for personalized medicine approaches that account for patient-specific microbiota compositions and metabolic profiles. Future research should prioritize the development of next-generation probiotics and engineered microbial strains capable of producing targeted neuroprotective and anti-inflammatory effects [167]. Additionally, large-scale, longitudinal clinical trials are essential to validate both the benefits and risks of these therapies in diverse populations [125,168].

In one study, FMT demonstrated neuroprotective effects in APPswe/PS1dE9 transgenic mice, significantly improving cognitive deficits, reducing Aβ accumulation, and alleviating synaptic dysfunction and neuroinflammation [167]. These protective effects may be associated with the restoration of gut microbiota composition and its metabolites, particularly short-chain fatty acids (SCFAs) [167]. Despite these findings, there is currently insufficient evidence to support the routine use of FMT as a treatment for AD.

### 5.4. Additional Therapeutic Options

Therapeutic strategies targeting both LPS and gingipains, such as small molecule inhibitors or IgY antibody-containing lozenges against gingipains, represent additional potential treatment avenues [125]. These options, though intriguing, require further research and clinical validation for their effectiveness in AD management [169].

## 6. Conclusions

This review analyzes the evidence that AD and its development depend on certain interrelated factors. These factors include iron dysregulation, intestinal and oral dysbiosis (which cause local inflammation, leading to the translocation of bacterial metabolites and inflammation in the nervous system), inflammation at a systemic level, cellular deterioration (resulting from the beta-amyloid accumulation and neurofibrillary tangles), abnormalities in blood vessel function and iron regulation, as well as digestive disorders and disruptions in oral microbiota. The evidence published supports the notion that AD is associated with and brought about by dysregulated inflammatory factors circulating in the body. Iron is continuously deposited in the brain with aging, leading to iron dysregulation and potential oxidative stress. Iron homeostasis dysregulation produces neurotoxic substances and reactive oxygen species, causing iron-induced oxidative stress. Bacteria-produced inflammatory agents can infiltrate the body via two distinct routes: either through the gut or through a two-step process that starts in the oral cavity and proceeds through the bloodstream to the brain, leading to neuroinflammation. At this stage of the disease, pathological changes initially impacting the olfactory system, progress to the temporal lobe, and eventually extend to the brainstem.

These processes are potential targets for therapeutic intervention. Several treatments have demonstrated potential for treating AD (including prebiotics, probiotics, antibiotics, and fecal transplant treatments). At present, the primary obstacle in drug development aimed at treating intricate cognitive illnesses, including AD, is the pursuit of multimodal drugs that can alter the course of the disease. Efforts to identify the root causes of AD, such as gut dysbiosis and iron toxicity, have resulted in the creation of diverse and innovative treatments. These treatments not only have the potential to alleviate the loss of motor control seen in AD but can also significantly decelerate the disease’s advancement.

This review uniquely integrates current evidence on the interplay between iron homeostasis dysregulation and microbial dysbiosis, emphasizing their collective role in oxidative stress, neuroinflammation, and neurodegeneration in AD. By presenting the dual impact of these mechanisms and highlighting novel therapeutic strategies such as probiotics, iron chelation, and fecal microbiota transplantation, this work offers a multidimensional perspective on potential targets for intervention. Furthermore, this review underscores the importance of adopting a personalized and multimodal therapeutic approach, bridging microbiological, biochemical, and clinical perspectives. This synthesis not only advances our understanding of AD pathogenesis but also provides actionable insights for future interdisciplinary research and innovation in treatment strategies.

To prevent AD and its progression, we propose pinpointing the origins of the illness and any accompanying health issues and then implementing a tailored treatment plan that emphasizes its core attributes. This personalized treatment plan should encompass a thorough investigation of all factors involved. To achieve this goal, it is essential to commence extensive longitudinal studies with sizable groups of individuals. Prospective tactics could center on biomarker utilization to predict AD advancement before blatant cognitive impairment emergence. This guideline aims to provide medication that can modify the trajectory of a disease in its early stages, typically before any symptoms manifest themselves. The intended outcome is to hinder or postpone disease onset. Ultimately, attempts to restore the gut flora in AD patients may significantly delay neurodegeneration by reducing inflammatory responses and amyloidogenesis.

To facilitate the clinical application of these therapies, several steps must be taken. First, standardized protocols for probiotics, prebiotics, and fecal microbiota transplantation (FMT) need to be established. These protocols should include precise dosages, optimal delivery methods, and long-term monitoring guidelines to ensure safety and efficacy. For example, tailoring probiotic regimens to individual microbiota profiles, identified through advanced sequencing techniques, may enhance their therapeutic potential.

Second, integrating biomarkers such as inflammatory cytokines, gut microbiota composition, and iron metabolism indicators into diagnostic workflows can improve patient stratification and treatment personalization. This approach enables early identification of individuals most likely to benefit from these interventions. Third, future clinical trials should focus on multimodal treatment regimens combining existing therapies, such as probiotics and iron chelation, with emerging options like engineered microbial strains producing neuroprotective metabolites. Conducting large-scale, longitudinal studies with diverse populations will provide critical data on efficacy, safety, and potential long-term outcomes. Lastly, establishing multidisciplinary research consortia can accelerate innovation by fostering collaboration between microbiologists, neurologists, and clinical practitioners. By addressing these factors, these therapies can transition from experimental treatments to standard care practices for AD.

While current therapeutic strategies such as probiotics, iron chelation therapy, antibiotics, and fecal microbiota transplantation (FMT) hold promise in addressing Alzheimer’s disease (AD) pathogenesis, they face significant limitations. Probiotics and prebiotics, although beneficial in modulating gut–brain axis communication, exhibit variability in effectiveness due to differences in individual microbiota compositions and the lack of standardized dosing protocols. Similarly, iron chelation therapies like deferiprone have shown potential in reducing iron-induced neurotoxicity but may inadvertently affect systemic iron balance, leading to anemia or other side effects. Antibiotics, while capable of modulating microbiota, risk disrupting beneficial microbial populations and contributing to antibiotic resistance. FMT, despite its potential to restore microbial diversity, lacks long-term safety data and remains constrained by ethical and logistical challenges.

To overcome these limitations, future research should focus on novel approaches, such as the development of tailored, next-generation probiotics and microbiota-targeted therapeutics designed to enhance gut–brain communication. Advances in synthetic biology could allow for the engineering of microbial strains that produce neuroprotective metabolites or secrete anti-inflammatory agents specifically targeting AD-related pathways. Additionally, combining existing treatments into multimodal regimens, such as integrating probiotics with anti-inflammatory drugs or antioxidants, may enhance their cumulative efficacy. Furthermore, personalized medicine approaches, leveraging biomarkers to tailor treatments to an individual’s specific microbiome and metabolic profile, could significantly improve outcomes. Finally, large-scale, longitudinal clinical trials are essential to validate these novel strategies and ensure their safety and efficacy in diverse populations.

Table 1 provides a summary of key findings, proposed mechanisms, and clinical implications associated with iron homeostasis dysregulation, oral and gut dysbiosis, systemic inflammation, and therapeutic strategies in Alzheimer’s disease.

## Figures and Tables

**Figure 1 microorganisms-13-00122-f001:**
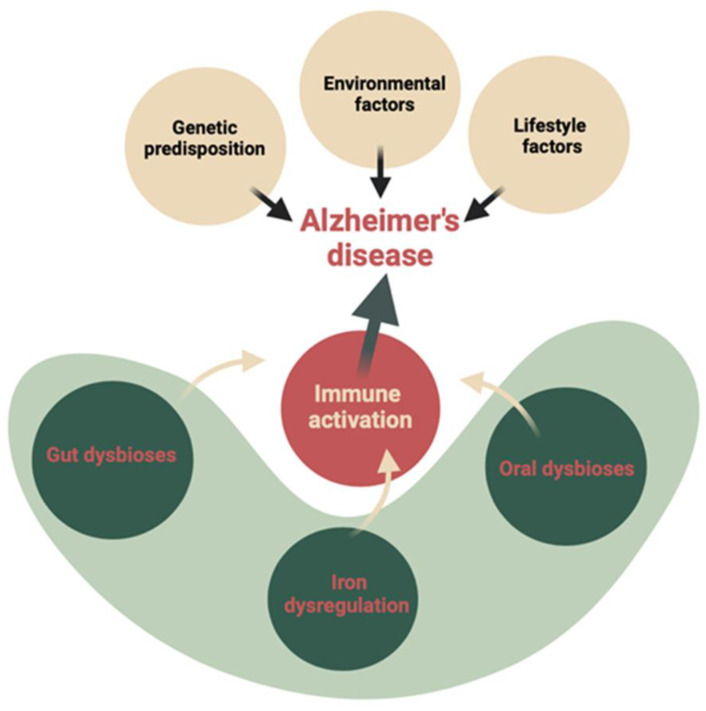
AD pathomechanisms.

**Figure 2 microorganisms-13-00122-f002:**
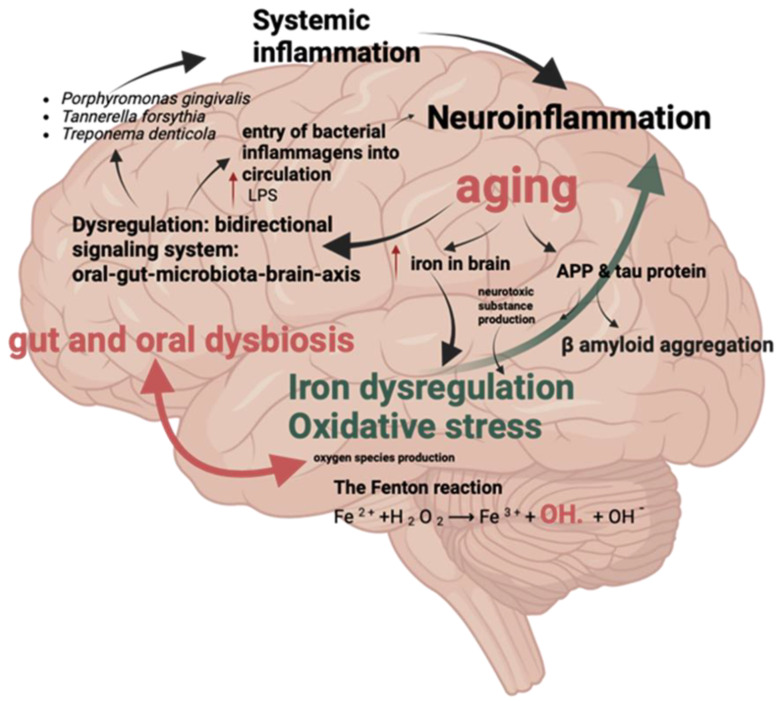
Iron dysregulation and inflammatory factors.

**Table 1 microorganisms-13-00122-t001:** Summary of key findings, mechanisms, and clinical implications.

Key Area	Research Findings	Proposed Mechanisms	Clinical Implications	References
Iron Homeostasis Dysregulation	Excess intracellular iron leads to ROS production, promoting oxidative stress and neuronal damage in AD.	Fenton reaction generates hydroxyl radicals; ferroptosis exacerbates neurodegeneration by lipid peroxidation.	Iron chelation therapy (e.g., deferoxamine) may reduce oxidative stress and protect neurons.	[11,23,47,54]
Oral and Gut Dysbiosis	Dysbiosis in oral (e.g., *P. gingivalis*) and gut microbiota associated with systemic inflammation and BBB disruption.	LPS translocation induces neuroinflammation; dysbiosis reduces beneficial SCFAs and increases pro-inflammatory cytokines.	Probiotics, prebiotics, and FMT can help restore microbial balance and reduce inflammation.	[18,108,123,167]
Systemic Inflammation	Elevated pro-inflammatory cytokines (e.g., IL-1β, IL-6, TNF-α) contribute to neuroinflammation and neuronal death.	Activation of microglia and astrocytes; cytokine-induced damage to neurons and promotion of amyloid-beta (Aβ) aggregation.	Anti-inflammatory therapies targeting cytokines (e.g., IL-6 inhibitors) can reduce neuroinflammation	[52,112,132,141]
Therapeutic Strategies	Emerging treatments include probiotics, antibiotics, iron chelation, and fecal microbiota transplantation (FMT).	Probiotics enhance gut–brain axis signaling and reduce inflammation; antibiotics modulate microbiota; iron chelation targets iron dysregulation; FMT restores microbiome composition.	Multimodal and personalized treatment approaches are needed to address the multifactorial nature of AD.	[44,58,162,167]

## Data Availability

Not applicable.

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
