# Peer review of "Iron Homeostasis Dysregulation, Oro-Gastrointestinal Microbial Inflammatory Factors, and Alzheimer’s Disease: A Narrative Review"

_microorganisms, 2025, doi:10.3390/microorganisms13010122_

Round 1
Reviewer 1 Report
Comments and Suggestions for Authors
Iron homeostasis dysregulation, oro-gastrointestinal microbial inflammatory factors, and Alzheimer's disease: a narrative review
Exciting review.
It would be best if you revised the abstract to cover the topic.
L45-48: revise
L286-292: revise, not clear
L295: remove however
Bacteria names should be in italics
L436: add a full stop after the reference.
L471: this? put the name
Author Response
Thank you for your valuable feedback and suggestions. We have carefully addressed each of the points raised:
Abstract Revision: The abstract has been revised to provide a clearer and more comprehensive summary of the manuscript's topic, emphasizing the key points and scope.
L45-48 Revision: This section has been rephrased for clarity and conciseness, ensuring it aligns with the overall narrative.
L286-292 Revision: The sentences in this segment have been rewritten to improve clarity and flow. We have clarified the intended meaning and ensured the arguments are coherent.
L295 – Removal of "However": The word "however" has been removed, and the sentence has been restructured for smoother readability.
Bacteria Names in Italics: All bacterial names throughout the manuscript have been checked and appropriately formatted in italics.
L436 – Full Stop Added: A full stop has been added after the reference to maintain proper punctuation.
L471 – Specify "This": The word "this" has been replaced with the specific name to avoid ambiguity.
Reviewer 2 Report
Comments and Suggestions for Authors
Review for the manuscript
Iron homeostasis dysregulation, oro-gastrointestinal microbial inflammatory factors, and Alzheimer's disease: a narrative review
Dear Editor,
I appreciate this invitation to review this manuscript that was submitted to Microorganisms.
Section:
Medical Microbiology
Topical Collection:
Feature Papers in Medical Microbiology
OVERALL COMMENTS
In this manuscript, based on the statement that “Alzheimer's disease (AD) is a common form of dementia and a neurodegenerative disorder that affects the nervous system and patients’ cognitive abilities, and current evidence implies that iron dysregulation and bacterial inflammatory factors may be associated with the oral and gut microbiome as critical factors in developing this condition. Due to this, the authors intended to review the current literature and present AD development interrelated with a complex series of co-factors affecting AD.
TITLE
The title is adequate for the study.
ABSTRACT
The abstract needs to be improved. Please include the aim of the study, the main results, and the conclusions in a way that can differentiate these sections from one another.
KEYWORDS
The keywords used were “Keywords: Alzheimer's disease; Iron dysregulation; Bacterial inflammatory factors, Oral microbiota, Dysbiosis, Neuroinflammation
I suggest: Keywords: Alzheimer's disease; Iron dysregulation; Bacterial inflammatory factors, Oral microbi- 21 ota, Dysbiosis, Neuroinflammation; Dementia
INTRODUCTION
This section needs some modifications.
1- Please include articles published in 2023 and 2024 so this study can be considered as an evaluation of the current literature (as stated by the authors at the end of this section);
2- Please see in lines 48-50, where we read that “This article analyzes the current literature and presents AD development interrelated with a complex series of co-factors affecting the condition”.
I suggest: “The aim of this study is to evaluate the relationship between AD development interrelated with a complex series of co-factors affecting the condition”. Please include this is a narrative review.
3- At the end of this section we find Figure 1. After its legend (in lines 59-620 we find “Description: This figure illustrates the fundamental factors contributing to Alzheimer's disease (AD) pathomechanisms. Recent studies have highlighted the impact of style/environment, iron dysregulation, and oral/intestinal dysbiosis on immune system activation during AD onset. These three factors directly influence the AD-characteristic brain neurodegeneration process.”
I suggest removing this description and the legend of Figure 1 be:
Figure 1. Fundamental factors contributing to Alzheimer's disease (AD) pathomechanisms. Recent studies have highlighted the impact of style/environment, iron dysregulation, and oral/intestinal dysbiosis on immune system activation during AD onset. These three factors directly influence the AD-characteristic brain neurodegeneration process.
DISCUSSION
In lines 373-377 we can read that “Figure 2. Iron Dysregulation and Inflammatory Factors.
Description: This figure shows the relationship between iron dysregulation and inflammatory factors associated with oral and gut microbiota, which are implicated in the development of Alzheimer's disease.”
I suggest removing the “description” and starting the legend of this Figure as:
Figure 2. Relationship between iron dysregulation and inflammatory factors associated with oral and gut microbiota, which are implicated in the development of Alzheimer's disease.
(Please expand this explanation for the information included in this Figure).
CONCLUSIONS
This section is fine.
LIMITATIONS and STRENGTHS
I suggest including the limitations and the strengths of this study.
FUTURE DIRECTIONS
I suggest including the Future directions of this study. What do the authors expect from the next steps?
REFERENCES
More articles should be published in 2024.
Author Response
Thank you very much for taking the time to review this manuscript. We appreciate your constructive feedback, which has helped improve the quality and clarity of our work. Below are detailed responses to each comment, with corresponding revisions highlighted in the re-submitted manuscript.
ABSTRACT
Comments: The abstract needs to be improved. Please include the aim of the study, the main results, and the conclusions in a way that can differentiate these sections from one another.
Response: Thank you for this valuable suggestion. We have revised the abstract to clearly distinguish the aim, main results, and conclusions. These changes are reflected in the updated abstract on [lines 11-21, updated text in the manuscript: Abstract: Alzheimer's disease (AD), the most common form of dementia, is a progressive neuro-degenerative disorder that profoundly impacts cognitive function and the nervous system. Emerging evidence highlights the pivotal roles of iron homeostasis dysregulation and microbial inflammatory factors in the oral and gut microbiome as potential contributors to the pathogenesis of AD. Disruption of iron homeostasis can result in excessive intracellular iron accumulation, promoting the generation of reactive oxygen species (ROS) and oxidative damage. Additionally, inflammatory agents produced by pathogenic bacteria may enter the body via two primary pathways: directly through the gut or indirectly via the oral cavity, entering the bloodstream and reaching the brain. This infiltration disrupts cellular homeostasis, induces neuroinflammation, and exacerbates AD-related pathology. Addressing these mechanisms through personalized treatment strategies that target the underlying causes of AD could play a critical role in prevent-ing its onset and progression.].
Comments: The keywords used were: "Keywords: Alzheimer's disease; Iron dysregulation; Bacterial inflammatory factors, Oral microbiota, Dysbiosis, Neuroinflammation." I suggest: "Keywords: Alzheimer's disease; Iron dysregulation; Bacterial inflammatory factors, Oral microbiota, Dysbiosis, Neuroinflammation; Dementia."
Response: Thank you for this suggestion. We have updated the keywords to include "Dementia" for better indexing and relevance.
INTRODUCTION
Comments 2: Please see in lines 48-50, where we read: “This article analyzes the current literature and presents AD development interrelated with a complex series of co-factors affecting the condition.” I suggest: “The aim of this study is to evaluate the relationship between AD development interrelated with a complex series of co-factors affecting the condition.” Please include this is a narrative review.
Response 2: We agree with this suggestion and have updated the text to include the recommended phrasing.
Comments 3: At the end of this section we find Figure 1. After its legend (in lines 59-62), we find: “Description: This figure illustrates the fundamental factors contributing to Alzheimer's disease (AD) pathomechanisms. Recent studies have highlighted the impact of style/environment, iron dysregulation, and oral/intestinal dysbiosis on immune system activation during AD onset. These three factors directly influence the AD-characteristic brain neurodegeneration process.”
I suggest removing this description and the legend of Figure 1 be: Figure 1. Fundamental factors contributing to Alzheimer's disease (AD) pathomechanisms. Recent studies have highlighted the impact of style/environment, iron dysregulation, and oral/intestinal dysbiosis on immune system activation during AD onset. These three factors directly influence the AD-characteristic brain neurodegeneration process.
Response 3: Thank you for this suggestion. We have revised the legend for Figure 1 as recommended and removed the redundant description.
DISCUSSION
Comments: In lines 373-377, we read: “Figure 2. Iron Dysregulation and Inflammatory Factors. Description: This figure shows the relationship between iron dysregulation and inflammatory factors associated with oral and gut microbiota, which are implicated in the development of Alzheimer's disease.” I suggest removing the “description” and starting the legend of this Figure as: Figure 2. Relationship between iron dysregulation and inflammatory factors associated with oral and gut microbiota, which are implicated in the development of Alzheimer's disease. (Please expand this explanation for the information included in this Figure).
Response: Thank you for this helpful feedback. We have revised the legend for Figure 2 to remove the description and expand on the information provided.
Reviewer 3 Report
Comments and Suggestions for Authors
The manuscript "Iron Homeostasis Dysregulation, Oro-Gastrointestinal Microbial Inflammatory Factors, and Alzheimer's Disease: A Narrative Review" is scientifically sound and addresses an important topic. The manuscript addresses a novel and interdisciplinary topic by linking iron dysregulation, microbiota, and Alzheimer's Disease (AD). Provides an in-depth narrative review of complex mechanisms involving iron homeostasis, gut-brain interaction, and neuroinflammation. However, I have some minor comments to be considered.
- Narrow the scope by specifying whether the focus is on theoretical mechanisms, clinical evidence, or both.
- Provide a succinct statement of objectives in the introduction to set clearer boundaries for the review.
- Although the narrative highlights key links between microbiota, iron dysregulation, and AD, some mechanisms (e.g., ferroptosis) are underexplored in relation to their direct contribution to AD pathology.
- Discuss more directly how bacterial metabolites influence AD progression beyond general inflammatory pathways.
- Certain topics, such as the relationship between gut microbiota and systemic inflammation, are revisited multiple times, leading to repetitive content.
- Transitions between sections (e.g., from oral dysbiosis to therapeutic approaches) are abrupt and could be smoothed for better readability.
- Consider adding a summary table, consolidating key research findings, mechanisms, and clinical implications.
- Explicitly highlight how this review contributes uniquely to the field.
- Discuss potential limitations of current therapeutic strategies and propose novel directions.
- Discuss potential risks of therapies (e.g., FMT-induced microbiota shifts) alongside benefits.
- Provide more specific recommendations for integrating these therapies into clinical practice or future trials.
Author Response
Thank you very much for taking the time to review this manuscript. We appreciate your constructive feedback, which has helped improve the quality and clarity of our work. Below are detailed responses to each comment, with corresponding revisions highlighted in the re-submitted manuscript.
Comments 1: Narrow the scope by specifying whether the focus is on theoretical mechanisms, clinical evidence, or both.
Response 1: Thank you for this insightful comment. We agree with this suggestion and have clarified in the introduction that the review focuses on both theoretical mechanisms and clinical evidence. The statement of scope can now be found on [lines 62–66, updated text in the manuscript: To further specify the scope of this review, our focus is primarily on the theoretical mechanisms underlying the interplay between iron dysregulation and microbial in-flammatory factors in Alzheimer’s disease. We additionally incorporate selected clinical evidence to contextualize and support these mechanisms, providing a comprehensive but targeted analysis.]
Comments 2: Provide a succinct statement of objectives in the introduction to set clearer boundaries for the review.
Response 2: Thank you for pointing out this opportunity for improvement. We have now added a concise objective statement in the introduction to set clear boundaries for the review. This addition can be found on [56-61, updated text in the manuscript: The aim of this review is to present the current knowledge on the relationship between the development of AD and various interrelated factors including iron homeostasis dysregulation, iron-induced oxidative stress and dysbiosis in the oral and gut microbiota leading to immune activation and neuroinflammationin the brain as well as to highlight therapeutic options in AD patients such as iron chelation therapy, probiotics and antibiotics administration, and fecal microbiota transplantation].
Comments 3: Although the narrative highlights key links between microbiota, iron dysregulation, and AD, some mechanisms (e.g., ferroptosis) are underexplored in relation to their direct contribution to AD pathology.
Response 3: We appreciate this observation. Additional discussion on ferroptosis and its direct contribution to AD pathology has been incorporated in the section [2.2], which can be found on [149-166, updated text in the manuscript:Ferroptosis, a distinct iron-dependent form of regulated cell death, plays a significant role in AD pathology through lipid peroxidation and ROS generation [50, 51]. This process, characterized by the depletion of glutathione and the inactivation of glutathione pe-roxidase 4 (GPX4), leads to the accumulation of toxic lipid peroxides that compromise neuronal integrity [53]. Elevated levels of serum ferritin and iron observed in AD patients correlate with cognitive decline, suggesting that ferroptosis is not merely a secondary phenomenon but a contributing factor to neurodegeneration [54]. Furthermore, ferrop-tosis promotes amyloid-beta aggregation and tau hyperphosphorylation, two hallmark features of AD. Dysregulated iron metabolism exacerbates oxidative stress and in-flammation, initiating a vicious cycle that accelerates neuronal loss [56]. Emerging studies propose that targeting ferroptosis through iron chelation and the use of ferroptosis in-hibitors, such as liproxstatin-1 and deferoxamine, may offer therapeutic benefits in mitigating AD progression [45, 164]. Despite these findings, the specific molecular pathways linking ferroptosis to AD remain underexplored. Future research should focus on identifying biomarkers that can accurately measure ferroptosis activity in vivo, as well as developing targeted interventions that disrupt this pathological cascade [55]. Inte-grating ferroptosis inhibitors into multimodal treatment strategies could offer a novel avenue for reducing neuronal damage in AD [167, 171].].
Comments 4: Discuss more directly how bacterial metabolites influence AD progression beyond general inflammatory pathways.
Response 4: Thank you for this suggestion. We have expanded the discussion on bacterial metabolites and their influence on AD progression, specifically detailing mechanisms beyond general inflammatory pathways. This updated content can be found on [535-582, updated text in the manuscript : 4.6 Direct Mechanisms of Bacterial Metabolites in AD Pathogenesis. Bacterial metabolites influence AD progression by targeting specific molecular pathways beyond general inflammation. SCFAs, such as butyrate, regulate the expres-sion of anti-inflammatory cytokines and enhance BBB integrity, mitigating neuro-degeneration. Meanwhile, tryptophan metabolites like quinolinic acid, derived from the kynurenine pathway, exacerbate excitotoxicity and oxidative stress, directly contributing to neuronal death. Bacterial amyloid-like proteins, such as curli, promote amyloid-beta aggregation in a prion-like manner, accelerating plaque formation and protein mis-folding. Additionally, LPS disrupt the BBB and activate microglia, amplifying neuroin-flammation and facilitating tau hyperphosphorylation. Finally, microbial modulation of iron metabolism induces ferroptosis, an iron-dependent form of cell death linked to cognitive decline. Together, these mechanisms highlight how bacterial metabolites di-rectly contribute to AD pathogenesis, beyond systemic inflammation, emphasizing their role as potential therapeutic targets. 4.6.1 Amyloid Mimicry and Aggregation. Bacterial amyloid-like proteins, such as curli fibers, share structural similarities with human amyloid-beta (Aβ). These microbial proteins can act as nucleating agents, pro-moting Aβ aggregation in a prion-like manner. This interaction accelerates plaque formation and protein misfolding, key features of AD pathology. Furthermore, bacterial amyloids activate toll-like receptor 2 (TLR2), triggering neuroinflammation and oxidative stress [5, 99]. 4.6.2 Kynurenine Pathway and Tryptophan Metabolism. The bacterial modulation of tryptophan metabolism significantly impacts the kynurenine pathway, a crucial process in AD. Dysbiosis-induced shifts in this pathway lead to increased production of neurotoxic metabolites, such as quinolinic acid and 3-hydroxykynurenine. These compounds contribute to oxidative stress, excitotoxicity, and synaptic dysfunction—hallmarks of AD progression. Tryptophan depletion further im-pairs serotonin synthesis, exacerbating neuronal communication deficits and mood dis-turbances [152, 154]. 4.6.3 Iron Dysregulation and Ferroptosis. Bacteria influence systemic and local iron metabolism through siderophore pro-duction and interactions with host iron-binding proteins. This dysregulation facilitates iron accumulation in neural tissues, exacerbating oxidative stress via the Fenton reaction. The resulting lipid peroxidation and ferroptosis—a form of iron-dependent cell death—lead to neuronal damage and cognitive decline in AD [45, 50, 55]. 4.6.4 Neuroactive Metabolites and Neurotransmitter Modulation. Gut bacteria produce a variety of neuroactive compounds, including serotonin, dopamine, and gamma-aminobutyric acid (GABA). Dysbiosis disrupts the balance of these neurotransmitters, impairing synaptic function and cognitive processes. For ex-ample, reduced serotonin levels are linked to increased Aβ aggregation and neurotoxi-city, highlighting the critical role of the gut-brain axis in AD progression [68, 125]. 4.6.5 Lipopolysaccharides (LPS) and Peripheral Immune Activation. LPS compromises the blood-brain barrier (BBB), allowing peripheral inflammatory molecules to infiltrate neural tissues. This facilitates Aβ deposition and tau hyperphos-phorylation, compounding neurodegenerative processes. Additionally, LPS directly ac-tivates microglial cells, promoting chronic neuroinflammation and neuronal damage [110, 144, 145]].
Comments 6: Transitions between sections (e.g., from oral dysbiosis to therapeutic approaches) are abrupt and could be smoothed for better readability.
Response 6: Thank you for this helpful feedback. We have revised transitions between sections to improve flow and coherence. For instance, the transition between oral dysbiosis and therapeutic approaches now includes a linking paragraph summarizing how oral microbiota research informs therapeutic strategies. The updated text can be found on [412-415, updated text in the manuscript :Understanding the intricate connections between oral and gut dysbiosis and systemic inflammation provides a foundation for exploring targeted therapeutic strategies. Addressing these imbalances at their source may offer promising interventions to mitigate neurodegeneration in AD.]
Comments 7: Consider adding a summary table, consolidating key research findings, mechanisms, and clinical implications.
Response 7: We agree that a summary table would enhance clarity and readability. A new table summarizing key research findings, underlying mechanisms, and their clinical implications has been added to the manuscript as Table 1 on [page 18].
Comments 8: Explicitly highlight how this review contributes uniquely to the field.
Response 8: We appreciate this suggestion and have included a section in the conclusion emphasizing the unique contributions of this review to the field. This addition can be found on [714-724, updated text in the manuscript: This review uniquely integrates current evidence on the interplay between iron homeostasis dysregulation and microbial dysbiosis, emphasizing their collective role in oxidative stress, neuroinflammation, and neurodegeneration in Alzheimer’s disease (AD). By presenting the dual impact of these mechanisms and highlighting novel therapeutic strategies such as probiotics, iron chelation, and fecal microbiota transplan-tation, this work offers a multidimensional perspective on potential targets for inter-vention. Furthermore, the review underscores the importance of adopting a personalized and multimodal therapeutic approach, bridging microbiological, biochemical, and clin-ical perspectives. This synthesis not only advances our understanding of AD pathogenesis but also provides actionable insights for future interdisciplinary research and innovation in treatment strategies].
Comments 9: Discuss potential limitations of current therapeutic strategies and propose novel directions.
Response 9: Thank you for this valuable feedback. A new subsection discussing limitations of current therapeutic strategies and potential novel directions has been added under [Section 6], found on [755-779, updated text in the manuscript: While current therapeutic strategies such as probiotics, iron chelation therapy, an-tibiotics, and fecal microbiota transplantation (FMT) hold promise in addressing Alz-heimer’s disease (AD) pathogenesis, they face significant limitations. Probiotics and prebiotics, although beneficial in modulating gut-brain axis communication, exhibit variability in effectiveness due to differences in individual microbiota compositions and the lack of standardized dosing protocols. Similarly, iron chelation therapies like de-feriprone have shown potential in reducing iron-induced neurotoxicity but may inad-vertently affect systemic iron balance, leading to anemia or other side effects. Antibiotics, while capable of modulating microbiota, risk disrupting beneficial microbial populations and contributing to antibiotic resistance. FMT, despite its potential in restoring microbial diversity, lacks long-term safety data and remains constrained by ethical and logistical challenges. To overcome these limitations, future research should focus on novel approaches, such as the development of tailored, next-generation probiotics and microbiota-targeted therapeutics designed to enhance gut-brain communication. Advances in synthetic bi-ology could allow for the engineering of microbial strains that produce neuroprotective metabolites or secrete anti-inflammatory agents specifically targeting AD-related path-ways. Additionally, combining existing treatments into multimodal regimens, such as integrating probiotics with anti-inflammatory drugs or antioxidants, may enhance their cumulative efficacy. Furthermore, personalized medicine approaches, leveraging bi-omarkers to tailor treatments to an individual’s specific microbiome and metabolic profile, could significantly improve outcomes. Finally, large-scale, longitudinal clinical trials are essential to validate these novel strategies and ensure their safety and efficacy in diverse populations].
Comments 10: Discuss potential risks of therapies (e.g., FMT-induced microbiota shifts) alongside benefits.
Response 10: Thank you for raising this important point. We have included a discussion on the risks associated with microbiota-targeted therapies, such as fecal microbiota transplantation (FMT), alongside their potential benefits. This discussion can be found on [659-676, updated text in the manuscript: While fecal microbiota transplantation (FMT) shows promise in restoring microbial diversity and reducing neuroinflammation, its application carries potential risks. These include unintended shifts in microbiota composition, which may lead to dysbiosis or overgrowth of pathogenic strains, particularly in vulnerable populations such as elderly or immunocompromised individuals (e.g., [174, 175]). Moreover, the long-term safety and efficacy of FMT in addressing Alzheimer’s disease (AD) remain insufficiently explored, and ethical concerns regarding donor selection and procedure standardization persist [173]. Similarly, probiotics and prebiotics, while beneficial, can cause adverse effects such as bloating or metabolic disturbances, depending on individual microbiome variability [166, 168]. Antibiotics, though useful in modulating harmful bacteria, may lead to anti-biotic resistance or disruptions in beneficial microbiota populations, potentially worsening gut-brain axis integrity [171]. Despite these challenges, these therapeutic strategies highlight the need for personalized medicine approaches that account for patient-specific microbiota compositions and metabolic profiles. Future research should prioritize the development of next-generation probiotics and engineered microbial strains capable of producing targeted neuroprotective and anti-inflammatory effects [167, 170]. Addition-ally, large-scale, longitudinal clinical trials are essential to validate both the benefits and risks of these therapies in diverse populations [175, 176]].
Comments 11: Provide more specific recommendations for integrating these therapies into clinical practice or future trials.
Response 11: We have added specific recommendations for integrating microbiota-targeted therapies into clinical practice and future research trials. This addition is included in the conclusion and summarized in Table X. Changes can be found on [743-779, updated text in the manuscript: To prevent AD and its progression, we propose pinpointing the origins of the illness and any accompanying health issues and then implementing a tailored treatment plan that emphasizes its core attributes. This personalized treatment plan should encompass a thorough investigation of all factors involved. To achieve this goal, it is essential to commence extensive longitudinal studies with sizable groups of individuals. Prospective tactics could center on biomarker utilization to predict AD advancement before blatant cognitive impairment emergence. This guideline aims to provide medication that can modify the trajectory of a disease in its early stages, typically before any symptoms manifest themselves. The intended outcome is to hinder or postpone disease onset. Ul-timately, attempts to restore the gut flora in AD patients may significantly delay neu-rodegeneration by reducing inflammatory responses and amyloidogenesis. To facilitate the clinical application of these therapies, several steps must be taken. First, standardized protocols for probiotics, prebiotics, and fecal microbiota transplantation (FMT) need to be established. These protocols should include precise dosages, optimal delivery methods, and long-term monitoring guidelines to ensure safety and efficacy. For example, tailoring probiotic regimens to individual microbiota profiles, identified through advanced sequencing techniques, may enhance their therapeutic potential. Second, integrating biomarkers such as inflammatory cytokines, gut microbiota composition, and iron metabolism indicators into diagnostic workflows can improve pa-tient stratification and treatment personalization. This approach enables early identifi-cation of individuals most likely to benefit from these interventions. Third, future clinical trials should focus on multimodal treatment regimens combining existing therapies, such as probiotics and iron chelation, with emerging options like engineered microbial strains producing neuroprotective metabolites. Conducting large-scale, longitudinal studies with diverse populations will provide critical data on efficacy, safety, and potential long-term outcomes. Lastly, establishing multidisciplinary research consortia can accelerate inno-vation by fostering collaboration between microbiologists, neurologists, and clinical practitioners. By addressing these factors, these therapies can transition from experi-mental treatments to standard care practices for Alzheimer’s disease].